| **Open Peer Review** | Bacteriology | New-Data Letter

# rRNA expression kinetics during the chlamydial developmental cycle

Yuxuan Wang,[1] Joseph D. Fondell,[1] Guangming Zhong,[2] Huizhou Fan[1]

**KEYWORDS** *Chlamydia*, rRNA, ribosomes, gene expression

Ribosomes, complexes of three distinct rRNAs (namely, 16S, 23S, and 5S rRNAs) and more than 50 ribosomal proteins, are essential for bacterial protein synthesis (1). In free-living bacteria, ribosomes are more abundant when cultured with rich culture media and during the logarithmic growth phase (2).

*Chlamydia* is an obligate intracellular bacterium with a unique developmental cycle characterized by two contrasting cellular forms termed the elementary body (EB) and reticulate body (RB) (3). After invading host cells, EBs differentiate into proliferating RBs within 6–8 h. RBs differentiate back into EBs beginning approximately 24 h post-infection (hpi). The progeny EBs and residual RBs are released from host cells beginning at about 40 hpi (3).

RB proliferation requires abundant protein synthesis. Accordingly, RBs must have more ribosomes than EBs. In the *Chlamydia trachomatis* genome, rRNA genes are organized as a 16S, 23S, and 5S rRNA sequence within two identical rRNA operons (Fig. 1A) (4, 5). Cleavages within the primary transcript (pre-rRNA) produce the three mature rRNAs (Fig. 1A). We recently reported that during the first hour of *C. trachomatis* infection, pre-, 16S, and 23S rRNAs undergo significant upregulation, whereas 5S rRNA does not (6). Further analysis led to the discovery that 5S rRNA is present in excess relative to 16S and 23S rRNAs, indicating that ribosomes can increase in the absence of a concurrent 5S rRNA upregulation during the immediate early phase (6). Given the essential role of rRNAs in protein synthesis, as well as their use as loading controls in chlamydial gene expression studies, we expanded our rRNA expression analysis to encompass the entire developmental cycle.

We performed qRT-PCR analysis to determine the expression levels of rRNAs in *C. trachomatis* L2 (strain 434/BU) cultured in L929 cells (6). We employed three primer pairs, each amplifying sequences present in mature rRNAs and the pre-rRNA, and additional primer pair targeting only pre-rRNA (Fig. 1A). The expression levels of rRNAs at various developmental points relative to 0 hpi are presented in Fig. 1B. Notably, by 1 hpi, an early point in the EB-to-RB differentiation, the expression levels of pre-rRNA, 16S rRNA, and 23S rRNA surged by 180-fold, 2.1-fold, and 2.5-fold, respectively, compared to their baselines at 0 hpi (Fig. 1B). By the 3 hpi midpoint of primary differentiation, these levels further increased to 1,091-fold for pre-rRNA and 5.6-fold each for both 16S and 23S rRNAs (Fig. 1B). By 8 hpi, the onset of the logarithmic RB growth phase, the expression peaked at 5,243-fold for pre-rRNA, 59-fold for 16S rRNA, and 62-fold for 23S rRNA (Fig. 1B). Thereafter, the expression levels decreased progressively through 32 hpi and then stabilized through 40 hpi (Fig. 1B). 5S rRNA displayed an expression pattern similar to those of 16S and 23S rRNAs, but with a distinguishable feature: its level did not increase until 3 hpi (Fig. 1B).

Percentages of individual rRNAs out of the total (Fig. 1C) and copies of rRNAs per cell (Fig. 1D) both show that 5S rRNA is the most abundant rRNA at 0 and 1 hpi, suggesting

Address correspondence to Huizhou Fan, fanhu@rwjms.rutgers.edu.

The authors declare no conflict of interest.

See the funding table on p. 3.

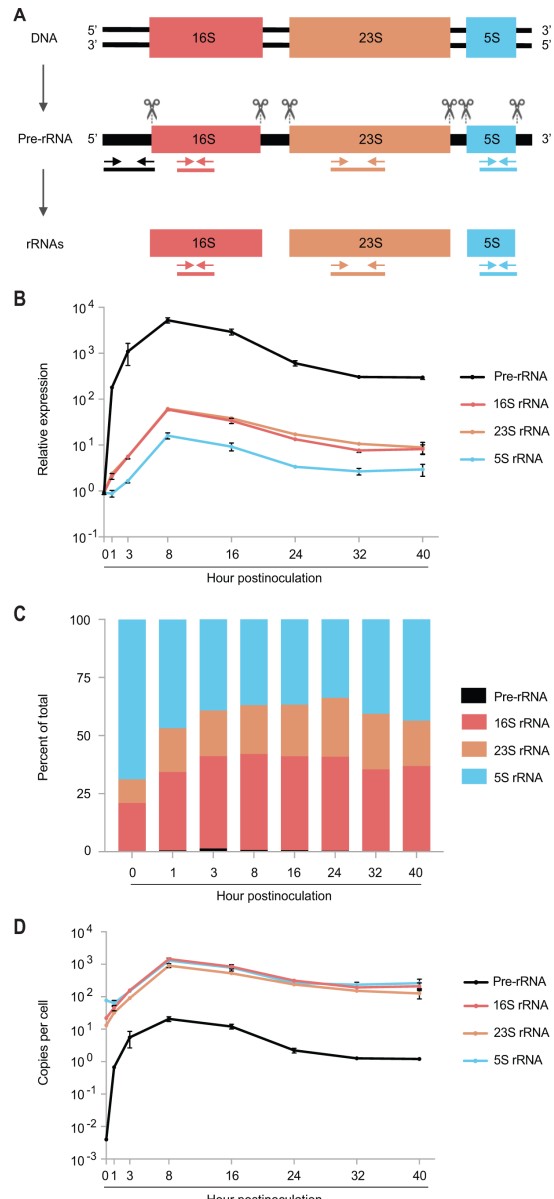

**FIG 1** Expression kinetics of pre- and mature rRNAs during the *C. trachomatis* developmental cycle. (A) Schematic presentation of *C. trachomatis* chromosomal rRNA gene organization and rRNA genesis. Scissor symbols indicate RNA cleavage at indicated sites. Arrows represent primers targeting specific sequences for amplifying pre-rRNA and mature rRNA. Lines below the arrows signify reverse transcription PCR products using different respective primer pairs. (A) (scheme of rRNA gene organization and rRNA maturation) was previously used in Wurihan et al. (6). (B) Temporal changes in rRNA levels from 0 to 40 hpi. Pre- and mature rRNAs were quantified via quantitative reverse transcription real-time PCR (qRT-PCR), utilizing primers indicated in (A). The expression level for each rRNA at 0 hpi was set at 1. Genome copy number data, obtained via qPCR, were used to normalize rRNA expression levels. (C) Proportional shifts of pre- and mature rRNAs from 0 to 40 hpi. qPCR was performed with genomic DNA as the template to determine the amplification efficiency of primer pairs targeting rRNAs. Proportions of individual rRNAs relative to the total rRNA pool were derived from the amplification efficiency-normalized copy numbers. (D) Copy numbers of rRNAs per cell from 0 to 40 hpi. (B, D) Data are averages ± standard deviations from biological triplicates.

that 5S rRNA carried over in EBs is slightly better preserved than 16S and 23S rRNAs. 23S rRNA remained the rate-limiting mature RNA throughout the developmental cycle. Fig. 1C and D also indicate that pre-rRNA maturation in *C. trachomatis* is highly efficient.

In summary, our observations demonstrate that the expression of rRNAs is intricately regulated in *Chlamydia*. Consequently, we recommend caution when using any rRNA as a loading control for chlamydial gene expression analysis.

## ACKNOWLEDGMENTS

This work was supported by grants from the National Institutes of Health (Grant # AI140167 and AI154305 to H.F.).

## AUTHOR AFFILIATIONS

[1]Department of Pharmacology, Robert Wood Johnson Medical School, Rutgers, The State University of New Jersey, Piscataway, New Jersey, USA
[2]Department of Microbiology and Immunology, University of Texas Health San Antonio, San Antonio, Texas, USA

## AUTHOR ORCIDs

Yuxuan Wang  http://orcid.org/0000-0003-0771-1078
Joseph D. Fondell  http://orcid.org/0000-0002-7046-9854
Guangming Zhong  http://orcid.org/0000-0001-7053-5009
Huizhou Fan  http://orcid.org/0000-0002-4903-926X

## FUNDING

| Funder | Grant(s) | Author(s) |
| --- | --- | --- |
| HHS | National Institutes of Health (NIH) | AI140167, AI154305 | Huizhou Fan |

## ADDITIONAL FILES

The following material is available online.

### Open Peer Review

**PEER REVIEW HISTORY (review-history.pdf).** An accounting of the reviewer comments and feedback.

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
