## [Reviewer comments · Microbiology Spectrum]

Microbiology Spectrum

rRNA Expression Kinetics during the Chlamydial Developmental Cycle

Yuxuan Wang, Joseph Fondell, and Huizhou Fan

Corresponding Author(s): Huizhou Fan, Rutgers Robert Wood Johnson Medical School

Review Timeline:

Submission Date:	February 1, 2024
Editorial Decision:	March 28, 2024
Revision Received:	April 16, 2024
Accepted:	May 14, 2024

Editor: Dhammika Navarathna

Reviewer(s): Disclosure of reviewer identity is with reference to reviewer comments included in decision letter(s). The following individuals involved in review of your submission have agreed to reveal their identity: David E Nelson (Reviewer #1)

Transaction Report:

DOI: <https://doi.org/10.1128/spectrum.00132-24>

Re: Spectrum00132-24 (rRNA Expression Kinetics during the Chlamydial Developmental Cycle)

Dear Prof. Huizhou Fan:

Thank you for the privilege of reviewing your work. Below you will find my comments, instructions from the Spectrum editorial office, and the reviewer comments.

Revision Guidelines

Sincerely,
Dharmika Navarathna
Editor
Microbiology Spectrum

Reviewer #1 (Comments for the Author):

This interesting note from Wang et al shows that it may not be safe to assume that levels of all chlamydial rRNAs are similar, although the key difference appears to be only in the 5S transcript which appears to survive carryover in EBs a little better than the other rRNAs (which could be conceptually interesting, but the thought is not developed well in this note). I think it makes a minor addition to the literature with the caveat that I think they do not present some of the data very clearly (see note re 1B) and separately, that I think it would be useful to the readers to see how the copy numbers of the individual rRNAs and pre-RNA

compare to genome copy numbers (which they) have in another panel (add a 1D). I think this might better illustrate their point that rRNA copy numbers vary a lot during chlamydial development.

Sentence starting line 22: Add "bacteria" so subject of this sentence is clear.

Line 24: Are we sure chlamydia is a coccus? Looks pretty polymorphic. If someone has made this argument based on the PBP complement I guess calling it a coccus would be OK.

Line 30: Re-write to make it clear that we would predict that RBs would contain more ribosomes, it is not clear as written.

Line 63: How about using the ratio of a specific rRNA (say 16S) to genome copy number?

Comments on figures

1B. The Y axis here is not rRNA/genome. It is the ratio of each of the rRNA at time 0 to the compared timepoint, correct?

1C. Enough detail is not provided to clarify how you came up with these numbers. I am assuming that, first, the Q-PCR counts were normalized to a clone copy of the allele in a plasmid or something. Then, the counts from each of the 4 Q-PCRs was added and proportions were derived from that? This confusing, because the label in 1B makes it look like there is a lot of the pre-RNA, when that is actually a fold change.

Reviewer #2 (Comments for the Author):

This is a very straightforward short report addressing rRNA abundance inside developing chlamydia. I have four notes for the authors to address.

1- I looked, and I could not find any other assessment of differential rRNA abundance in chlamydia developmental forms. Do any of the transcriptomics studies address this? Please indicate that this was explored.

2- Can a modest expansion of the significance be included? What does this mean for the biology of the organism? Are there parallels in other bacteria?

3- Line 24: "Gram-negative bacteria" is better than "coccus".

4- Line 62: I think it should be rRNA not rRNAs.

Response to Reviewers' Comments:

We thank the reviewers for carefully evaluating our work and appreciate their constructive criticisms. We have revised the manuscript by addressing their critiques. It is our belief that the revised manuscript represents a much improved work. Please find point-by-point responses (in blue text) to reviewers' comments (in black text) below. Please refer to the line numbers indicated in the TrackChanges version of the manuscript.

REVIEWER 1

1) This interesting note from Wang et al shows that it may not be safe to assume that levels of all chlamydial rRNAs are similar, although the key difference appears to be only in the 5S transcript which appears to survive carryover in EBs a little better than the other rRNAs (which could be conceptually interesting, but the thought is not developed well in this note). I think it makes a minor addition to the literature with the caveat that I think they do not present some of the data very clearly (see note re 1B) and separately, that I think it would be useful to the readers to see how the copy numbers of the individual rRNAs and pre-rRNA compare to genome copy numbers (which they have in another panel (add a 1D). I think this might better illustrate their point that rRNA copy numbers vary a lot during chlamydial development.

Response: Thank you for your insightful comments. In the revised manuscript, lines 73 – 75 now read “Percentages of individual rRNAs out of the total (Fig. 1C) and copies of rRNAs per cell (Fig. 1D) both show that 5S rRNA is the most abundant rRNA at both 0 and 1 hpi, suggesting that 5S rRNA carried over in EBs is slightly better preserved than 16S and 23S rRNAs.” To further illustrate your point regarding variability in rRNA copy numbers during the chlamydial developmental cycle, we have added Fig. 1D, which presents the copies of rRNAs per cell. This new panel very well conveys the notion that rRNA copy numbers vary greatly during the chlamydial developmental cycle.

2) Sentence starting line 22: Add "bacteria" so subject of this sentence is clear.

Response: In the revised manuscript, we have combined the last two sentence in paragraph 1 to meet the length requirement. The new sentence (lines 20-22) now reads “In free-living bacteria, ribosomes are more abundant when cultured with rich culture media and during the logarithmic growth phase”.

3) Line 24: Are we sure chlamydia is a coccus? Looks pretty polymorphic. If someone has made this argument based on the PBP complement I guess calling it a coccus would be OK.

Response: We are not aware of any chlamydial PBP complementation studies. In the revised manuscript, we have changed “coccus” to “bacterium”.

4) Line 30: Re-write to make it clear that we would predict that RBs would contain more ribosomes, it is not clear as written.

Response: Thank you for your feedback. In the revised manuscript, the third paragraph now starts with “RB proliferation requires abundant protein synthesis. Accordingly, RBs must have more ribosomes than EBs.”

5) Line 63: How about using the ratio of a specific rRNA (say 16S) to genome copy number?

Response: Please refer to the response to point 1.

6) Fig. 1B. The Y axis here is not rRNA/genome. It is the ratio of each of the rRNA at time 0 to the compared timepoint, correct?

Response: You are right! The original Y-axis label “rRNA/genome” was not accurate. In the revised manuscript, we have changed the Y-axis label to “Relative expression”.

7) Fig. 1C. Enough detail is not provided to clarify how you came up with these numbers. I am assuming that, first, the Q-PCR counts were normalized to a clone copy of the allele in a plasmid or something. Then, the counts from each of the 4 Q-PCRs was added and proportions were derived from that? This confusing, because the label in 1B makes it look like there is a lot of the pre-RNA, when that is actually a fold change.

Response: In the resubmitted manuscript, we have added “qPCR was performed with genomic DNA as the template to determine the amplification efficiency of primer pairs targeting rRNAs. Proportions of individual rRNAs relative to the total rRNA pool were then derived from the amplification efficiency-normalized copy numbers.” to the legend of Fig. 1C (lines 127-130).

REVIEWER 2

1. I looked, and I could not find any other assessment of differential rRNA abundance in chlamydia developmental forms. Do any of the transcriptomics studies address this? Please indicate that this was explored.

Response: We are not aware of any previous studies that have carefully assessed rRNA abundance in chlamydial developmental forms. In transcriptomics studies, rRNAs are usually removed before library construction to increase the coverage of mRNAs. Residual rRNA transcripts detected do not reflect their physiological levels.

2. Can a modest expansion of the significance be included? What does this mean for the biology of the organism? Are there parallels in other bacteria?

Response: We appreciate your suggestion to expand on the significance of our findings. Unfortunately, due to the strict 500-word limit for letters, we are unable to include a more extensive

discussion in this submission. However, we are actively investigating whether the phenomena observed in chlamydia are also present in other bacteria, and we plan to address these parallels in future publications.

3. Line 24: "Gram-negative bacteria" is better than "coccus".

Response: In the resubmitted manuscript, "Gram-negative coccus" has been changed to "Gram-negative bacterium".

4. Line 62: I think it should be rRNA not rRNAs.

Response: Thanks for your note. In the resubmitted manuscript, we have changed "rRNAs expression" to "the expression of rRNAs" line 78.

Re: Spectrum00132-24R1 (rRNA Expression Kinetics during the Chlamydial Developmental Cycle)

Dear Prof. Huizhou Fan:

Your manuscript has been accepted, and I am forwarding it to the ASM production staff for publication. Your paper will first be checked to make sure all elements meet the technical requirements. ASM staff will contact you if anything needs to be revised before copyediting and production can begin. Otherwise, you will be notified when your proofs are ready to be viewed.

Sincerely,
Dharmika Navarathna
Editor
Microbiology Spectrum

Reviewer #2 (Comments for the Author):

The revised manuscript addresses each of the modest concerns by previous reviewers.